# Metabolic Pathophysiology of Cortical Spreading Depression: A Review

**DOI:** 10.3390/brainsci14101026

**Published:** 2024-10-16

**Authors:** Arren Hill, Alfred B. Amendolara, Christina Small, Steve Cochancela Guzman, Devin Pfister, Kaitlyn McFarland, Marina Settelmayer, Scott Baker, Sean Donnelly, Andrew Payne, David Sant, John Kriak, Kyle B. Bills

**Affiliations:** Department of Biomedical Science, Noorda College of Osteopathic Medicine, Provo, UT 84606, USA; do25.adhill@noordacom.org (A.H.); casmall@noordacom.org (C.S.); do25.sbguzmanc@noordacom.org (S.C.G.); do25.dppfister@noordacom.org (D.P.); do25.kamcfarland@noordacom.org (K.M.); do25.mmsettelmayer@noordacom.org (M.S.); do27.scbaker@noordacom.org (S.B.); do25.smdonnelly@noordacom.org (S.D.); ajpayne@noordacom.org (A.P.); dwsant@noordacom.org (D.S.); jakriak@noordacom.org (J.K.); kbbills@noordacom.org (K.B.B.)

**Keywords:** cortical spreading depression, migraine, traumatic brain injury, biomarkers

## Abstract

Cortical spreading depression (CSD) is an electrophysiologic pathological state in which a wave of depolarization in the cerebral cortex is followed by the suppression of spontaneous neuronal activity. This transient spread of neuronal depolarization on the surface of the cortex is the hallmark of CSD. Numerous investigations have demonstrated that transmembrane ion transport, astrocytic ion clearing and fatigue, glucose metabolism, the presence of certain genetic markers, point mutations, and the expression of the enzyme responsible for the production of various arachidonic acid derivatives that participate in the inflammatory response, namely, cyclooxygenase (COX), all influence CSD. Here, we explore the associations between CSD occurrence in the cortex and various factors, including how CSD is related to migraines, how the glucose state affects CSD, the effect of TBI and its relationship with CSD and glucose metabolism, how different markers can be measured to determine the severity of CSD, and possible connections to oligemia, orexin, and leptin.

## 1. Introduction

Cortical spreading depression (CSD) is a pathological state in which a wave of depolarization in the cerebral cortex is followed by the suppression of spontaneous neuronal activity [1]. Although its clinical relevance is widely accepted, comprehensive etiological explanations are lacking. Recent studies have revealed numerous associations between CSD and other pathological and physiological processes, including migraines, glucose dysregulation, traumatic brain injury (TBI), cerebrovascular accidents (CVA), various biomarkers, and oligemia, orexin, and leptin [2,3]. 

Owing to the complexity of CSD, its role in neurological injury and disease has not yet been completely characterized. Nevertheless, CSD is a crucial aspect of some neurological pathologies and represents a potential target for monitoring and intervention that warrants further investigation. This paper presents the existing literature exploring the different factors involved in cortical spreading depression episodes and how these factors may be linked to glucose dysregulation, migraines, and other neurological pathologies.

## 2. Literature Review 

### 2.1. Cortical Spreading Depression Is a Complex Pathological Process 

Numerous inciting events have been linked to CSD. The existing literature shows spontaneously observed CSD propagation, as well as experimentally reproducible CSDs, caused by TBIs, CVAs, migraines, dysglycemic states, excessive stimulation, increased oxygen demand, and chemical insults [4,5,6]. There has also been an observed link between stress on the body and factors such as nutrition, mental stress, sleep, age, alcohol, and inflammatory states that may modulate susceptibility to CSD [7]. While the events that can precipitate CSD are numerous, they share a common pathologic alteration of the neuronal homeostatic balance. The underlying connection is often some type of triggering event that causes cellular injury, leading to an alteration in functioning, and increasing the susceptibility of the neuronal tissue to CSD propagation (Figure 1). CSD readily occurs after these insults or inciting events.

Physiologically, CSD is characterized by an influx of sodium and calcium ions in concentrations that overwhelm the sodium pump’s transport capacity [8]. This results in sustained neuronal depolarization, producing a slowly spreading wave that propagates further depolarization in other cortical regions. This leads to disrupted ion concentrations that cannot be readily corrected. Thus, neurons are placed in a state of increased energy demand and energy production attempting to correct the ion imbalance [9,10,11]. 

With the sodium-potassium ATPase at the cellular level no longer able to maintain sustained neuronal firing and depolarization, the extracellular matrix is left with an overwhelming buildup of potassium, a hallmark finding of CSD [12]. Similarly, mutations in the Na-K-ATPase are found in spreading depression phenotypes of migraines, indicating the key role of this pump in CSD [13,14]. Potassium homeostasis in the brain is also tightly regulated by processes such as astrocytic gap junctions, which contribute to spatial buffering. When potassium regulation fails, events such as CSD, anoxic depolarization, and epileptiform activity may be precipitated [15]. 

With the extracellular matrix in a hyperkalemic state, surrounding neuronal tissue adjacent to the origin of the CSD is more readily depolarized. Hyperkalemia shifts the resting membrane potential closer to the threshold of depolarization as these changes in ionic concentrations alter adjacent voltage-gated channels on these nearby neurons [16]. The resulting microenvironment enables the spread of CSD outward from the original source. This process disrupts neuronal transmembrane ion gradients, and in the setting of the previously mentioned neurological injuries, can trigger and propagate episodes of CSD. The resulting alterations to synaptic structures cause further downstream complications, such as changes in vascular responses and the subsequent depression of electrical activity, which may exacerbate injury and lead to neuronal death [17,18]. 

While the stepwise progression of CSD can be detailed in turn, the exact mechanisms underlying CSD, from the inciting event to resolution, are not completely understood. The depolarization wave includes electrolyte movement of a standard depolarization followed by an inability to correct electrolyte concentrations, namely, an efflux of potassium ions and an influx of sodium and calcium. This wave of depolarization, which results in a dysregulated electrolyte concentration, is driven and spread by a downstream efflux of excitatory amino acids, including glutamate and aspartate. Owing to electrolyte imbalance and the dependence on Na-K-ATPase to correct this electrolyte imbalance, neurons are left in a state of increased energy metabolism while also being in a state of variable cerebral blood flow [17]. 

Astrocytes also aid in the clearance of K^+^ and glutamate in this electrolyte imbalance. However, in late CSD, the energy demand of increased K^+^ and glutamate clearance exceeds capacity, and Ca^2+^ signaling is abnormally hyperactive. This leads to a large-scale release of lactic acid, hydropic and endfoot swelling, and hypoxia, which further increases neuroinflammation and slows glymphatic flow (Figure 2) [19]. Following ion disequilibrium, the end state of CSD results in measurable changes in gene expression, as shown by increases in the expression of IL-6, IL-1beta, and TNFα [20]. 

Throughout the course of a CSD episode, CSD also disrupts normal cerebral hemodynamics in rats by affecting blood flow to brain tissue [21]. There have been three different phases of alterations to the normal cerebral hemodynamics that have been observed. These three alterations are hypoperfusion seen as the initial change after the induction of CSD, followed by an episode of profound hyperemia, and then concluding with oligemia and hypoxemia being seen post-CSD [22]. Alterations in cerebral hemodynamics have innumerable risks to cerebral tissue and are both a consequence and a conditional factor to CSD.

Cortical spreading depression has been observed to occur spontaneously in hypoxic, ischemic, or hypoglycemic brain tissue and can be induced experimentally [17]. For example, CSD has been observed following CVA, where the spreading depression first develops in the ischemic core and then spreads to the peri-infarct zone, where it may contribute to secondary injury and penumbra cell death. Studies have also associated the presence of CSD in the peri-infarct zone with increased infarct volume [23]. Additionally, CSD may be observed in 50–60% of patients with TBI. Damaged brain tissue following severe head injury creates a cellular environment that is more susceptible to CSD [24]. Moreover, CSD has been implicated in the pathophysiology of migraine headaches [25]. 

### 2.2. Variable Glycemic States Modulate Cortical Spreading Depression 

Varying glycemic states may have a direct influence on the onset of CSD. Experimentally, hyperglycemia has been shown to increase the electrical spreading depression threshold and reduce the frequency of these spreading depressions. This effect was observed when blood glucose levels were elevated to 400 mg/dL through dextrose infusion with CSD induced by the continuous topical application of potassium chloride (KCl). In contrast, hypoglycemia, induced by lowering blood glucose levels to 40 mg/dL via insulin, significantly prolonged both the frequency of individual spreading depression events and the cumulative duration of spreading depressions [26]. These characteristics of hypoglycemia and its direct effects on spreading depression have also been observed in states of nutritional deficiency and malnutrition [27]. 

A connection between hippocampal signaling firing, the glycemic state, and CSD has also been described in the literature. The hippocampus is one of the most epileptogenic areas in the brain and is suspected to be a trigger zone for hypoglycemic seizures [28,29]. Florez et al. demonstrated this experimentally by inducing hypoglycemia in mouse hippocampal tissue slices. They reported an alteration in the intrinsic CA3 hippocampal rhythms exhibiting decreased inhibition and increased excitation [30]. They also noted that spreading depression occurred following seizure-like events. As CSD progresses, some neuronal firing propagates into the hippocampus, with the entorhinal cortex playing a major role in its connection to subcortical structures [31]. These spreading depression-like events originated in two different regions, with the majority originating in the CA1 hippocampal region, three-quarters of which displayed irreversible synaptic failure, and the rest originated in the CA3 hippocampal region [30]. The induction of a hypoglycemic state and subsequent spreading depression-like events suggest a link between CSD onset and hypoglycemia.

The ventromedial hypothalamic nucleus (VHN) is known for its role in glucose sensing and its ability to influence whole-body glucose homeostasis [32]. Alterations in VHN activity have been observed when glucose levels deviate from the normal range. Prolonged hyperglycemia has been shown to increase the levels of brain-derived neurotropic factor (BDNF), and hypoglycemia has been shown to cause the overexpression of thioredoxin-1 in the VHN [33,34]. BDNF is highly expressed in all regions of the hypothalamus, has a role in neuronal survival, has functions that regulate glucose and energy metabolism at the cellular level, and prevents the exhaustion of pancreatic ß cells during prolonged states of hyperglycemia [35,36]. Thioredoxine-1 plays a role in attenuating oxidative stress and is neuroprotective during these hypoglycemic periods, during which oxidative stress is increased [37]. In the setting of CSD, acute hyperglycemia was shown to decrease cerebral blood flow in a stepwise fashion, implying that hyperglycemia may exacerbate cerebral damage after CSD caused by ischemic stroke [38]. Further, it has been shown that increased metabolic stress, including those caused by glucose fluctuations, when combined with oxidative stress, can exacerbate susceptibility to CSD [1,39].

### 2.3. Leptin Provides Insights into the Link between CSD, Migraines, and Metabolism 

Peptides such as insulin, glucagon, and leptin, which are involved in blood glucose and appetite regulation, could influence the neurobiology of migraines [40]. Leptin, a 16 kDa adipocytokine, is produced primarily by adipose tissue but is also synthesized in other tissues, including the brain, muscle, bone marrow, and stomach. Leptin has a variety of functions, including appetite suppression, energy homeostasis regulation, and the modulation of inflammatory and immune processes. Leptin’s association with CSD was observed in rats that were given leptin injections via intracerebroventricular administration. After administration, the number of CSDs that occurred increased significantly [41]. Leptin is also believed to play a significant role in migraine pathophysiology [42]. Researchers have reported that in states where the body has increased leptin levels, these subjects experienced an increase in the number of migraine attacks [43]. 

Leptin receptors are found in large numbers in the arcuate nucleus and dorsomedial hypothalamus. Studies have shown that individuals with migraines often exhibit sensitivity to fasting and skipping meals, which can trigger migraine attacks [43,44,45]. Understanding how diets regulate these hormones and their relationship with migraines may help link them to CSD.

Recent genetic studies suggest that migraines may be related to metabolic disorders, as elevations in free fatty acids and ketone bodies often precede a migraine attack [46,47]. Studies have also shown that there is a relationship between insulin resistance and migraines, as well as a correlation between increased glucose levels and higher rates of migraines [39,48,49]. The mechanisms by which increases and decreases in cerebral glucose contribute to the onset and duration of CSD remain poorly understood and is an area in need of further investigation. A potential neurobiological connection between migraines and compromised metabolic homeostasis could involve dysregulated blood glucose regulation [50]. It has been suggested that insulin, glucagon, and leptin may alter the transmission of nociceptive inputs from the trigeminal nerve to higher brain regions, implying that adipokine signaling could influence specific neural networks involved in the development of migraines [40].

### 2.4. Traumatic Brain Injury May Precipitate CSD through Changes in Glucose Metabolism 

Following TBI, there is a rapid, indiscriminate release of excitatory neurotransmitters, leading to ionic disequilibrium, much like that observed in the CSD, across neuronal membranes. Although a link between TBI and CSD has been recognized, a comprehensive understanding has some gaps, with glucose regulation potentially playing a role [51]. TBI is known to cause a state of altered glucose metabolism [52]. This glucose dysregulation may serve to further explain, at least in part, the underlying connection of TBI and CSD with the possible connection of these conditions to migraines.

Research has shown that excitatory stimuli resulting from the modulation of glutamate levels are elevated following head injury in patients [53]. This excitatory stimulus is how the CSD spreads out from its inciting source. After this wave of depolarization fires, there is also the mechanical disruption of the cell membrane, allowing an efflux of intracellular potassium through voltage-gated channels [54]. The process of returning to ionic homeostasis requires an increase in energy consumption via Na-K-ATPase in an attempt to return the ion concentration balance, which leads to an increase in demand for glucose usage [55]. However, approximately 5.7 days after TBI, brain oxygen consumption and glucose uptake are decreased [56]. Seizures are often associated with TBI, which may, in part, be a result of glucose dysregulation [56,57]. For example, decreased cellular oxygen and glucose uptake and consumption were observed following TBI.

After TBI, the body’s management of glucose is further disrupted as gluconeogenesis is altered. Lactate tracing from injured tissues following TBI revealed that 67.1% of glucose originates from gluconeogenesis from lactate, whereas 15.2% originates from gluconeogenesis in healthy patients, further revealing the relationship between neurological conditions and the dysregulation of glucose in the body [56].

In addition to TBI being associated with cortical spreading depression, CSD has also been linked to many other nontraumatic brain injuries, including aneurismal subarachnoid hemorrhages, delayed ischemic stroke after a subarachnoid hemorrhage, malignant hemispheric stroke, and a spontaneous intracerebral hemorrhage [58]. This highlights the significant overlap and interplay between these disease states.

### 2.5. C-Fos, COX-2, HO-1, and PKCδ Are Elevated in CSD 

Four markers have been identified in patients demonstrating CSD: c-Fos, cyclooxygenase-2 (COX-2), heme oxygenase-1 (HO-1), and protein kinase C-delta (PKCδ) [59,60,61]. The expression levels of these markers vary depending on the glycemic state.

During an episode of spreading depression, c-Fos and COX-2 are considered early response genes, whereas HO-1 and PKCδ are considered late response genes. Cortical spreading depression induced by the topical application of KCl increases the expression of each of these proteins [62]. When hypo- or hyperglycemic states were induced for 30 min, followed by the induction of spreading depression, the gene expression profile differed from that of CSD alone. In the CSD control normoglycemic state, CSD increased the gene expression of c-Fos by 340%, COX-2 by 210%, HO-1 by 470%, and PKCδ by 410%. In the case of hypoglycemia, c-Fos induction was enhanced by 145%, but HO-1 and PKCδ induction was reduced to 43% and 64%, respectively. In contrast, in the case of hyperglycemia, c-Fos induction was enhanced by 388% and COX-2 induction by 53%, but HO-1 and PKCδ induction was reduced by 54% and 51%, respectively [48]. 

While both glycemic states increase c-Fos expression, this increase is more pronounced in hyperglycemic conditions. Hyperglycemia increases the expression of both c-Fos and COX-2, which are early response genes, whereas both glycemic states decrease the expression of HO-1 and PKCδ, which are late response genes. Notably, HO-1 and PKCδ are expressed at the highest levels in normoglycemic animals. Additionally, hypoglycemia alone does not induce gene expression, but hyperglycemia alone increases c-Fos expression by 42% [48]. 

The above findings indicate that the level of expression of these genes is altered by different glycemic states following an episode of CSD. Given their altered expression, these genes could serve as markers to track disease progression or severity. Whether these markers are merely indicative of CSD or actively contribute to its pathophysiological process remains uncertain. While the supporting literature connects c-Fos and COX-2 to CSD, the relationships between CSD and HO-1 or PKCδ are less studied.

#### 2.5.1. Increased C-Fos May Be Correlated with CSD 

C-Fos is a proto-oncogene that is expressed in some neurons following depolarization and is typically found in small quantities in healthy subjects. It was first observed after seizure activity and noxious stimulation exposure in the spinal cord [63]. Since its discovery, various factors have been shown to influence c-Fos expression in addition to noxious stimuli. For example, levels of c-Fos are increased after exposure to general mechanical stimulation [63,64]. CSD has also been shown to increase c-Fos levels in the magnocellular region of the hypothalamic paraventricular nucleus [65]. Additionally, c-Fos is activated by transient cerebral ischemia and is believed to play a role in ischemic tolerance and cell survival [66]. Interestingly, predisposing cells to hyperglycemia and then subjecting them to ischemia without the induction of CSD completely blocked the expression of c-Fos. This finding suggests that c-Fos may be a predictor or even a contributor to hyperglycemia-enhanced ischemic brain damage [66]. 

Although thought to be helpful for cell survival, c-Fos may play a role in the pathophysiology of exacerbating CSD through its overexpression. Elevated levels of c-Fos have been implicated in migraine pathophysiology, particularly in familial hemiplegic migraine 2, which results from a point mutation (E700K) in the ATP1A2 exon. Compared with wild-type mice, mice with this mutation were shown to have a significantly faster propagation velocity and longer full width at half the maximum of CSD, as well as a lower threshold for CSD initiation. This may be caused by increased numbers of c-Fos-positive cells in the ipsilateral somatosensory cortex, piriform cortex, amygdala, and striatum and by a significant increase in c-Fos-positive cells in the ipsilateral amygdala of the mutated type animals compared with those in the wild-type animals. Higher c-Fos expression in the amygdala may indicate alterations in the limbic system, suggesting an enhanced linkage between cortical spreading depression and amygdala connectivity in familial hemiplegic migraine 2 patients [67]. 

#### 2.5.2. Altered Expression of COX-1 and -2 May Modulate CSD 

COX-2 is an enzyme that plays a substantial role in mediating inflammation. It is tightly regulated but easily induced and upregulated early during states of inflammation and ischemia [68]. One of the hallmarks of CSD is the development of persistent cortical oligemia, which is a reduction in blood volume circulating in an area and puts cells at risk of ischemia.

It has been proposed that this persistent oligemia may be caused by the presence of vasoconstricting eicosanoids [69]. Using laser Doppler flowmetry in urethane-anesthetized rats, it was determined that the oligemia response following CSD can be prevented. Administration of the nonselective COX inhibitor naproxen completely inhibited the oligemic response, selective (cyclooxygenase-1) COX-1 inhibition with SC-560 preferentially reduced the early reduction in CBF, and selective COX-2 inhibition with NS-398 affected only the later response. Moreover, blocking the action of thromboxane A2 (TXA2), a vasoconstriction eicosanoid, with ozagrel prevents the initial decrease in CBF, whereas the inhibition of prostaglandin F2α, another eicosanoid, via the administration of AL-8810 inhibits a later phase of oligemia [69]. 

Given the evidence that the glycemic state can alter the expression of COX during CSD, a connection between the glycemic state and COX level, which plays a role in modulating CSD, is possible. Another study demonstrated that the inhibition of COX-2 with celecoxib had promising results in reducing the severity of CSD and easing the intensity of migraine attacks. Celecoxib was shown to significantly reduce CSD-induced dilation of dural arteries as well as reduce the activation of dural and pial macrophages [70]. COX-2 may play a role in the disease process of CSD, as shown by its ability to block the induction or alter the course of CSD. However, the exact mechanism behind this phenomenon remains unknown. 

#### 2.5.3. The Roles of PKCδ and HO-1 in CSD Are Poorly Defined 

PKCδ is a protein kinase expressed ubiquitously among cells and plays a critical role in multiple cellular functions via phosphorylation, such as regulating cellular growth, differentiation, and apoptosis [71]. HO-1 is involved in cellular antioxidant defenses and antiapoptotic functions and belongs to the heat shock protein family [72]. As mentioned above, both PKCδ and HO-1 levels are altered during different glycemic states and following the induction of CSD, with their expression decreasing when CSD occurs in both hypo- and hyperglycemic states. 

PKCδ is known to be activated during ischemia [73]. Since protein phosphorylation is a fundamental mechanism involved in the regulation of most cellular processes, its activation during CSD is thought to be neuroprotective [74]. One study showed that blocking PKCδ expression by treating cells with an N-methyl-D-aspartate receptor antagonist, dizocilpine maleate, prevented the propagation of depression through cortical spreading [75]. However, additional evidence is lacking, and further research is needed to better understand this connection. Given the role of phosphorylation in numerous cellular functions, a better understanding of how PKC relates to or possibly perpetuates CSD is important.

### 2.6. CSD May Contribute to the Pathophysiology of Migraines 

The pathophysiology of migraines is vast and complex. The development and severity of migraines are related to various phenomena, including CSD, observed during a migraine attack, along with the dilation of intracranial blood vessels, activation of meningeal nociceptors, and activation of the trigemino-vascular pathway (Figure 3). The current literature strongly supports that the activation of the trigemino-vascular pathway is a core proponent of what perpetuates a migraine attack [76]. The trigemino-vascular system has neurons that project to a variety of areas in the brain, many of which are known to be areas associated with the symptomatic manifestations of migraines. This process is believed to involve the activation of these neurons by vasoactive peptides such as substance P, calcitonin gene-related peptide (CGRP), and pituitary adenylate cyclase-activating polypeptide (PACAP) [77]. 

The relationship between CSD and migraines is a topic of ongoing debate. The idea that CSD propagates migraine dates back to 1944 when CSD was first observed by Leão [78]. Since then, it has been proposed that this connection is associated with the efflux of potassium [25,78]. During states of elevated extracellular potassium concentrations, there is an increase in the release of CGRP, a major mediator of migraines [25]. Calcitonin gene-related peptide is known to be heavily involved in migraines for three major reasons: CGRP levels are elevated in the jugular outflow during migraine attacks, the intravenous injection of CGRP has been shown to directly cause migraines, and CGRP-based therapeutics are effective in treating migraines [79]. A potential interplay between CSD and migraines may occur through their respective relationship with CGRP illustrated by increased levels of CGRP and an increase in the cell size of CGRP mRNA-synthesizing neurons in the trigeminal ganglion following an episode of CSD [25].

Migraines have also been linked to altered glucose regulation, as shown by increased plasma glucose levels during migraine attacks [27]. This complicates the understanding of the underlying pathophysiology, as both migraines and CSD are linked to altered glycemic states. Migraines impact glucose metabolism in the brain, leading to cellular dysregulation, as shown by an increase in neuronal activation-to-resting glucose uptake ratio in the visual cortex of migraine patients between attacks. These findings suggest that glucose dysregulation is involved in multiple disease processes in the brain [56,57].

There is also an association between migraines and certain adipokines, such as leptin. The pathological process that leads to migraines involves the sensitization of the trigemino-vascular system, release of inflammatory markers, and initiation of a meningeal-like inflammatory reaction that is perceived as a headache [80]. The trigeminal nerve innervates the meninges, which are densely innervated by pain fibers. When inflammatory molecules, which are released during and after an episode of CSD, reach the meninges, an aura and, later, a painful headache may develop. Researchers who discovered this relationship suggested that this link may be the underlying mechanism of some neurological deficits in individuals with migraines with aura [28]. Aura is described by focal neurological experiences that are temporary and most often precede a migraine headache. Neurological experiences can be various symptoms from visual, sensory, speech, and/or motor symptoms, with visual being the most common, but migraine headaches can occur both with and without an aura phase [28]. Recent studies have suggested that CSD plays a critical role in the underlying mechanism behind migraines with aura [28].

It is believed that the brainstem and midbrain structures both play a role in either generating or perpetuating migraine processes [81]. It has also been suggested that neurons in the hypothalamus can activate meningeal nociceptors by altering the balance of parasympathetic and sympathetic tone in favor of parasympathetic tone, which may contribute to migraine pathogenesis [82]. Hypothalamic neurons may play a role in the regulation and firing of preganglionic parasympathetic neurons in the superior salivatory nucleus (SSN). The proposed imbalance is then propagated as the SNN can stimulate the release of acetylcholine (Ach), nitric oxide (NO), and vasoactive intestinal peptide from the sphenopalatine ganglion (SPG). This release of neurotransmitters and peptides causes intracranial blood vessel dilation, the extravasation of plasma proteins, and the release of inflammatory molecules that activate meningeal nociceptors. This pathway, as a mechanism for migraines, is supported by the experimental inhibition of SPG, which results in partial-to-complete relief of migraine symptoms [83].

## 3. Discussion 

Cortical spreading depression is associated with many different pathologies involving the brain, including migraines, glucose dysregulation, traumatic brain injury, cerebrovascular accidents, epilepsy, hemorrhages, and various biomarkers. Despite extensive research, the precise mechanisms by which these factors influence CSD have yet to be established (Figure 4).

Current research suggests that alterations in the glycemic state significantly impact the frequency, duration, and progression of CSD episodes. Both hypo- and hyperglycemic states can influence the frequency, duration, and progression of CSD episodes, yet the underlying mechanisms remain unclear. The current literature indicates a correlation between different glucose levels and CSD, which may be mediated by various chemical pathway messengers. However, the crucial question of how alterations in the expression of these markers modify the pathophysiology of CSD remains unanswered. A more comprehensive understanding of the mechanisms of these markers and their involvement in CSD could pave the way for a better understanding of how different glycemic states influence CSD, given their evident influence on the gene expression of these markers.

Additional research into the mechanisms underlying alterations in glucose levels and their role in influencing CSD is essential because a wide variety of circumstances lead to altered glucose physiologic conditions in the brain. It is well established that hypoglycemic states can trigger the onset of CSD, and research has shown that TBIs can affect the brain’s typical glucose usage. Understanding how these altered glucose conditions affect CSD could offer valuable insights into its pathophysiology.

The relationships among blood glucose levels, migraines, and the occurrence and progression of CSD are related. Although studies on the connection between migraines and glucose dysregulation are inconsistent, some suggest a bidirectional link between migraine, insulin resistance, and type two diabetes. This highlights the need for more comprehensive studies to clarify these associations.

Additionally, markers such as c-Fos, COX-2, HO-1, and PKCδ may clarify the effects of the glycemic state on CSD and the broader role of glucose in its pathophysiology. These markers have been shown to be elevated during a CSD event and may play a role in the progression of CSD. For instance, CSD severity is increased in mice overexpressing c-Fos, whereas the inhibition of COX has been shown to prevent the pathophysiology of the events involved in CSD. Although c-Fos and COX-2 are well documented in CSD pathophysiology, the literature surrounding CSD and its interplay between HO-1 and PKCδ is lacking, and further research is needed. Additionally, other markers, such as junB, c-jun, MKP-1, and hsp71, are also suspected to influence CSD, though their roles are outside the scope of this review.

The potential impact of these markers on CSD modulation becomes even more apparent when we consider the role of leptin and orexin. As CSD extends into the hippocampus, disrupting the intrinsic CA3 hippocampal rhythms, any molecule active in these brain regions, including leptin and orexin, could exert a significant influence on the overall modulation of CSD.

## 4. Conclusions 

Cortical spreading depression has complex relationships with, among other pathologies, migraines, TBI, and CVA. Despite significant existing work, the precise mechanism of interaction between these pathologies has not been fully elucidated and no unifying pathway has been presented. However, the recent literature indicates that both hypo- and hyperglycemic states can influence the frequency and progression of CSD episodes. Changes in neuronal glucose metabolism may provide a bridge between the involvement of CSD in migraines, TBI, and CVA as all these diseases impact glucose metabolism in the brain. Perhaps most interesting, migraines may both precede and be preceded by episodes of CSD. Additionally, markers such as c-Fos, COX-2, HO-1, and PKCδ may provide measurable indicators of CSD and further reveal interactions with neurological and metabolic pathologies.

### Future Directions 

With a better understanding of the interplay between these physiological processes, it may become possible to develop targeted therapies to prevent or halt CSD propagation, offering potential breakthroughs in protecting neuronal cells during conditions associated with CSD. Such advancements could lead to improved treatments and outcomes for individuals suffering from conditions linked to CSD, such as migraines and traumatic brain injuries. Thus, efforts should be made to answer standing questions surrounding the mechanism of CSD. Notably, the biomarkers discussed in this review require deeper investigation into the molecular pathways that precipitate their increase and provide a potential avenue for further research. 

## Figures and Tables

**Figure 1 brainsci-14-01026-f001:**
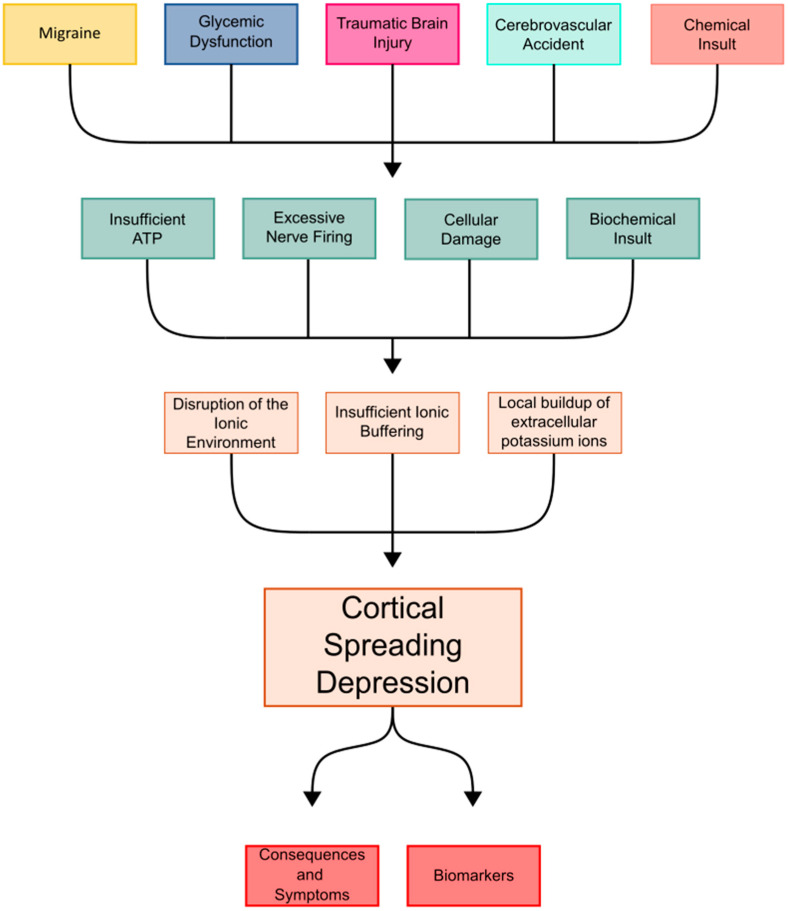
Flowchart describing the current understanding of CSD.

**Figure 2 brainsci-14-01026-f002:**
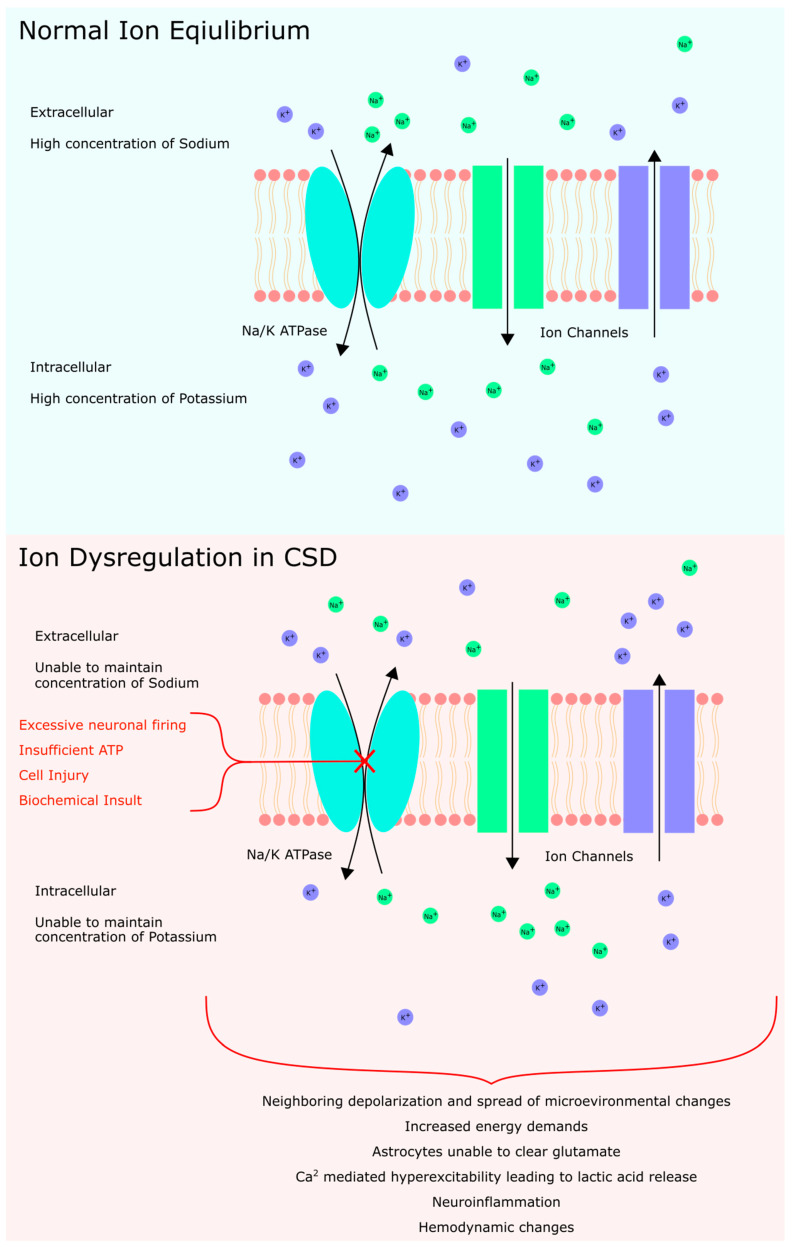
Summary of ionic disequilibrium resulting in CSD.

**Figure 3 brainsci-14-01026-f003:**
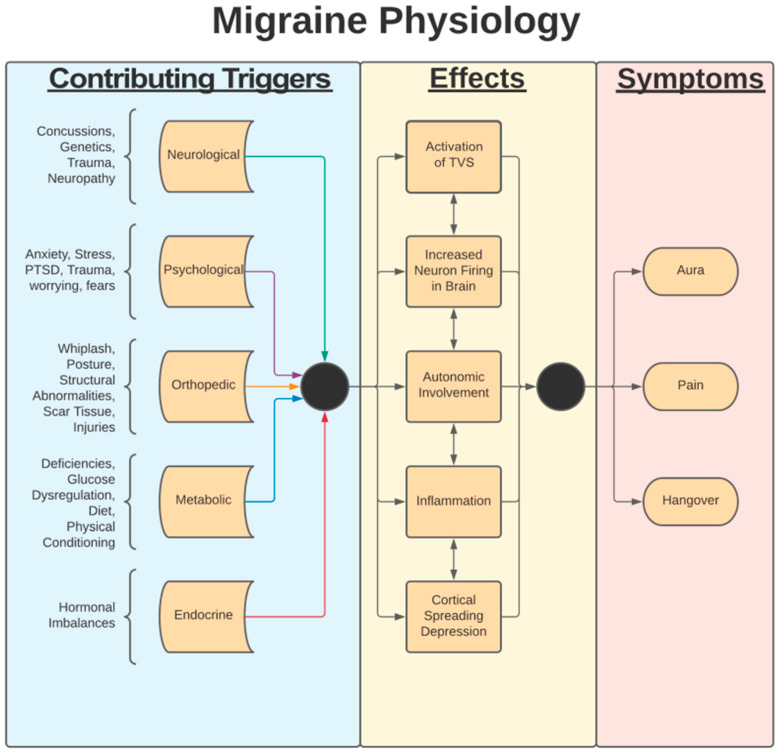
Flowchart of migraine physiology. Contributing triggers and migraine causes are not well understood.

**Figure 4 brainsci-14-01026-f004:**
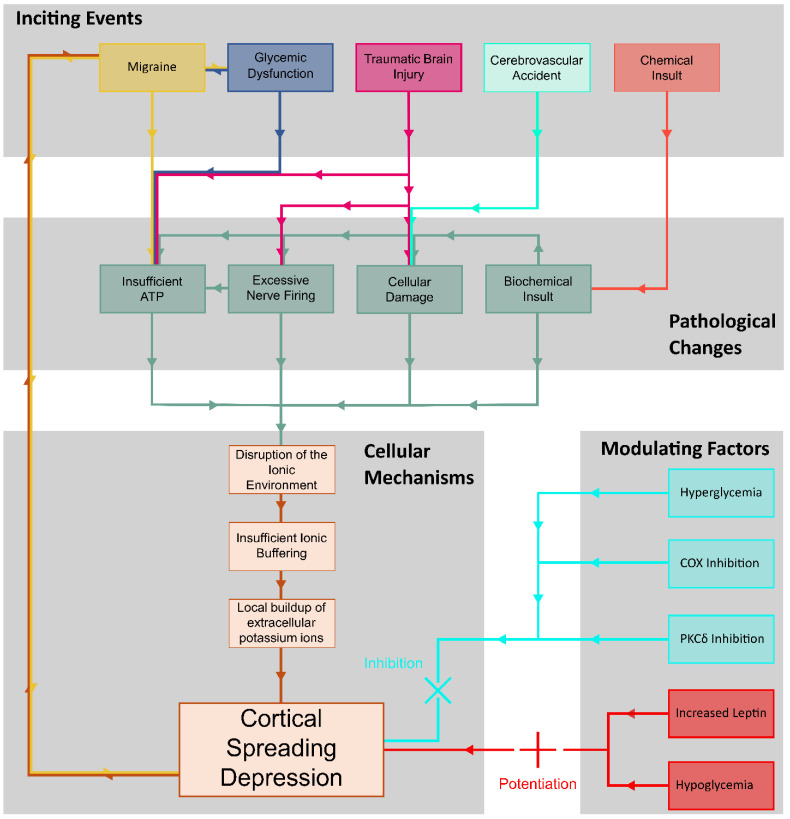
Summary diagram of CSD pathology and mechanism.

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
