# Peer review of "Metabolic Pathophysiology of Cortical Spreading Depression: A Review"

_brainsci, 2024, doi:10.3390/brainsci14101026_

Round 1
Reviewer 1 Report
Comments and Suggestions for Authors
This is a comprehensive, well-written review of the pathophysiological mechanisms of CSD. However, it mainly focuses on its metabolic aspects. This should be clarified in the title.
I have some more comments:
- A figure with ion influx/efflux under physiological conditions and CSD and its consequence (i.e., summarizing paragraph 2.1) would be highly explicative of the complicated processes.
- A paragraph focusing on the cerebral hemodynamic changes occurring during CSD, also discussing the relationship between CGRP and CSD, should be added
- Figure 2 should be modified. It gives the reader the wrong conception that migraine has some causes while it is a primary disorder. There is NO CAUSE-EFFECT mechanism. More specifically, the “orthopedic causes” should not be even mentioned.
Author Response
Comment 1: This is a comprehensive, well-written review of the pathophysiological mechanisms of CSD. However, it mainly focuses on its metabolic aspects. This should be clarified in the title.
Response 1: Thank you for your suggestion, we have elected to modify the title to: Metabolic Pathophysiology of Cortical Spreading Depression: A Review
Comment 2: A figure with ion influx/efflux under physiological conditions and CSD and its consequence (i.e., summarizing paragraph 2.1) would be highly explicative of the complicated processes.
Response 2: Thank you for your suggestion. We have included an additional figure to visualize the normal physiologic state vs the movement of ions in CSD. This is the new Figure 2.
Comment 3: A paragraph focusing on the cerebral hemodynamic changes occurring during CSD, also discussing the relationship between CGRP and CSD, should be added
Response 3: Thank you for your comment. We have added an expanded discussion of hemodynamic changes that occur with CSD – see lines 95-102. We have also added additional information about the direct relationship between CGRP and CSD. See lines 316-323 and 336-339.
Comment 4: Figure 2 should be modified. It gives the reader the wrong conception that migraine has some causes while it is a primary disorder. There is NO CAUSE-EFFECT mechanism. More specifically, the “orthopedic causes” should not be even mentioned.
Response 4: Thank you. We agree that the wording of figure 2 is somewhat inaccurate. Migraine is certainly classified as a primary disorder. We did not mean to imply a linear cause and effect relationship between the listed factors and the onset or diagnosis of migraine headache. Instead, our intent was to provide a list of commonly reported triggers or contributors for migraine headaches. The heading in figure 2 has therefore been changed from “Causes” to “Contributing Triggers”. It would likely be most accurate to say that the “causes” of migraine are not well understood and certainly multifactorial and though the items listed are well documented to be contributory triggers, the complexity of each of their roles in the causation is not understood.
We have elected to retain the “orthopedic” section as there is evidence in the literature that these factors may at least provoke migraine attacks – though no direct mechanistic explanation exists than we can identify. See the below reference as an example:
Weiss HD, Stern BJ, Goldberg J. Post-traumatic migraine: chronic migraine precipitated by minor head or neck trauma. Headache. 1991 Jul;31(7):451-6. doi: 10.1111/j.1526-4610.1991.hed3107451.x. PMID: 1774160.

Reviewer 2 Report
Comments and Suggestions for Authors
The authors introduced what is CSD and how CSD is linked to neurological diseases mainly migraine and TBI. They highlighted how glucose state affects CSD, and discussed the impact of CSD hallmarks on migraine and TBI. The paper adds new insights into CSD as a key pathophysiological event in these diseases. Yet, there are some areas of improvement:
Line12: The authors stated that this transient spread of neuronal depolarization on the surface of the cortex is the hallmark of CSD. CSD also involves astrocytes, particularly when propagating astrocytic calcium waves is linked to CSD. The involvement of astrocytes should be also considered when introducing CSD.
Line 40: Typots: reproduceable should be corrected
Line 57: When introducing sodium‒potassium ATPase, energy metabolism in CSD and its involvement in migraine conditions, it is suggested to include key references such as https://pubmed.ncbi.nlm.nih.gov/14586018/ by Reiffurth Clemens et al., 2019 CBFM, and https://pubmed.ncbi.nlm.nih.gov/30819023/ by Unekawa Miyuki et al., 2018 and extend the explanation on the role of energy metabolism in these conditions.
Line 102-104: ‘hypoglycemia with lower glucose prolonged both the duration of individual spreading depression events and the cumulative duration of spreading depressions [21]. Did the authors meant to state that increased cerebral glucose availability makes the tissue more resistant to SD’ thus possibly lower risks of having migraine? Please clarify
Line 153-154: The authors introduced how glucose state affects CSD and migraine risks and stated that there is correlation between increased glucose levels and higher rates of migraines [42, 43]. Do they show positive or negative correlation? Please specify. Does it mean increased glucose availability makes the tissue being more vulnerable to CSD? Looking at figure 3, it suggests that hypoglycemia promotes CSD.
Line 121-133: hyperglycemia increases the levels of BDNF, and hypoglycemia has been shown to cause the overexpression of thiore-doxin-1 in the VHN [28, 29]. How do ROS and glucose levels collectively affect CSD and increases migraine risks? Are there any relevant publications?
Line 191: ‘Four markers have been identified in patients demonstrating CSD: C-Fos, COX-2, HO-1, and PKCδ’. Actually, there are more than 4 molecules are equally important and identified after CSD. Key such molecules (such as CGRP, VIP) are all linked to BTI and migraine, which should be also briefly explained.
Line 194-217. The authors introduced how glucose state affects gene expression of C-Fos, COX-2, HO-1, and PKCδ induced by CSD to different extent. Are their proteins levels altered in these conditions? Is balancing glycemic statues by levigating these proteins a potential good strategy for reducing migraine/TBI progression?
Line 293:301. Is there any correlation between levels of plasma glucose and CGRP post CSD and/or migraine attacks?
Figure 2: Activation of TVS pathway as key pathophysiological event in migraine is currently missing in figure 2, which should be included and briefly introduced in the paper.
Author Response
Comment 1:
Line 12: The authors stated that this transient spread of neuronal depolarization on the surface of the cortex is the hallmark of CSD. CSD also involves astrocytes, particularly when propagating astrocytic calcium waves is linked to CSD. The involvement of astrocytes should be also considered when introducing CSD.
Response 1:
Thank you. We agree that astrocytes play an important role in the propagation of CSD, especially in the increased hypoxia and neuroinflammation in late-CSD. Astrocyte involvement is now introduced in the abstract (line 14). The role of astrocytes in early and late CSD is now summarized during a discussion of ion imbalances in lines 84-89, along with an appropriate reference.
Comment 2:
Line 40: Typos: "reproduceable" should be corrected.
Response 2:
Thank you. We have corrected this and edited the paper for any other typos/errors.
Comment 3:
Line 57: When introducing sodium‒potassium ATPase, energy metabolism in CSD, and its involvement in migraine conditions, it is suggested to include key references such as https://pubmed.ncbi.nlm.nih.gov/14586018/ by Reiffurth Clemens et al., 2019 CBFM, and https://pubmed.ncbi.nlm.nih.gov/30819023/ by Unekawa Miyuki et al., 2018, and extend the explanation on the role of energy metabolism in these conditions.
Response 3:
Thank you for those suggestions of references. They were added to lines 61-63 with added discussion of the relevance of the Na-K-ATPase in CSD based on the findings from these papers.
Comment 4:
Line 102-104: ‘Hypoglycemia with lower glucose prolonged both the duration of individual spreading depression events and the cumulative duration of spreading depressions [21].’ Did the authors mean to state that increased cerebral glucose availability makes the tissue more resistant to SD, thus possibly lowering the risk of having migraines? Please clarify.
Response 4:
Thank you for requesting clarification of this critical point. The text has been updated to provide further clarity that increased glucose availability increases the threshold for electrical spreading events while hypoglycemic states increase both the potential frequency and duration of these events.
Comment 5:
Line 153-154: The authors introduced how glucose state affects CSD and migraine risks and stated that there is a correlation between increased glucose levels and higher rates of migraines [42, 43]. Do they show positive or negative correlation? Please specify. Does it mean increased glucose availability makes the tissue more vulnerable to CSD? Looking at figure 3, it suggests that hypoglycemia promotes CSD.
Response 5:
Thank you, this is a needed clarification. Additional text has been added: “The mechanisms by which increases and decreases in cerebral glucose contribute to the onset and duration of CSD remain poorly understood and are an area in need of further investigation.” Additionally, another reference has been added to better cite the hyperglycemic influence on increased CSD.
Comment 6:
Line 121-133: Hyperglycemia increases the levels of BDNF, and hypoglycemia has been shown to cause the overexpression of thioredoxin-1 in the VHN [28, 29]. How do ROS and glucose levels collectively affect CSD and increase migraine risks? Are there any relevant publications?
Response 6:
The following has been added to the text for clarification: “Further, it has been shown that increased metabolic stress, including those caused by glucose fluctuations, when combined with oxidative stress, can exacerbate susceptibility to CSD [Busija 2008 and Charles and Brennan 2013].”
Comment 7:
Line 191: ‘Four markers have been identified in patients demonstrating CSD: C-Fos, COX-2, HO-1, and PKCδ.’ Actually, there are more than 4 molecules equally important and identified after CSD. Key molecules such as CGRP, VIP, etc., are all linked to BTI and migraine, which should also be briefly explained.
Response 7:
Thank you, the suggestion has been taken, and the role of CGRP and VIP has been added and expanded throughout the manuscript for greater accuracy and comprehensiveness.
Comment 8:
Line 194-217: The authors introduced how glucose state affects gene expression of C-Fos, COX-2, HO-1, and PKCδ induced by CSD to different extents. Are their protein levels altered in these conditions? Is balancing glycemic status by targeting these proteins a potential strategy for reducing migraine/TBI progression?
Response 8:
This is an excellent thought, and we agree that this should be the topic of future studies.
Comment 9:
Line 293-301: Is there any correlation between levels of plasma glucose and CGRP post-CSD and/or migraine attacks?
Response 9:
This is another excellent line of inquiry. We are currently unaware of any studies that have directly measured the link between plasma glucose, CGRP, and the onset of migraine attacks.
Comment 10:
Figure 2: Activation of the TVS pathway as a key pathophysiological event in migraine is currently missing in Figure 2, which should be included and briefly introduced in the paper.
Response 10:
Thank you for your comment. We have added this to Figure 2 and additionally have added several sentences discussing this specifically in section 2.6.

Reviewer 3 Report
Comments and Suggestions for Authors
This literature review provides a comprehensive overview of cortical spreading depression (CSD) and its links to various neurological conditions, particularly migraines and traumatic brain injury (TBI).
However, it could benefit from greater clarity and conciseness. Some sections are overly detailed, making it difficult to grasp the ultimate connections between concepts without delving deeper into the text. For instance, some sentences are long and complex, which can make it difficult for readers to grasp the main points. Breaking up lengthy sentences and using simpler structures could enhance readability.
There is some redundancy in discussing the effects of hypo- and hyperglycemic states on CSD. The idea that both extremes of glucose levels affect CSD frequency and duration is reiterated multiple times without adding substantial new information. Streamlining these points could help maintain reader engagement.
It is highly recommended to change the title of the manuscript, as it mainly focuses on the relationship between CSD and glycemic states.
The text makes broad claims about relationships between migraines, glucose dysregulation, and CSD without citing specific studies or evidence. Including references to relevant literature would strengthen the arguments and provide a foundation for the claims made (Sleep Med Rev. 2021 Jan 29;59:101449).
Additionally, while it highlights many relationships, it could more explicitly outline gaps in current research and the implications of these findings for clinical practice.
The conclusion touches on several important relationships but lacks a coherent synthesis of how these concepts interrelate. A more structured summary that emphasizes the connections between glycemic states, biomarkers, and CSD in the context of migraine, TBI, and CVA would provide a clearer takeaway for the reader.
Comments on the Quality of English LanguageSome sections are overly detailed, making it difficult to grasp the ultimate connections between concepts without delving deeper into the text. For instance, some sentences are long and complex, which can make it difficult for readers to grasp the main points. Breaking up lengthy sentences and using simpler structures could enhance readability.
Author Response
Comment 1: This literature review provides a comprehensive overview of cortical spreading depression (CSD) and its links to various neurological conditions, particularly migraines and traumatic brain injury (TBI).
However, it could benefit from greater clarity and conciseness. Some sections are overly detailed, making it difficult to grasp the ultimate connections between concepts without delving deeper into the text. For instance, some sentences are long and complex, which can make it difficult for readers to grasp the main points. Breaking up lengthy sentences and using simpler structures could enhance readability.
Response 1: Thank you for this feedback. We are unsure of which sections are overly detailed. This is meant to be a reasonably comprehensive review and thus we feel it would be counterproductive to reduce the depth of the material or the specificity with which it has been presented. We have, however, elected to edit the paper for clarity and conciseness.
Comment 2: There is some redundancy in discussing the effects of hypo- and hyperglycemic states on CSD. The idea that both extremes of glucose levels affect CSD frequency and duration is reiterated multiple times without adding substantial new information. Streamlining these points could help maintain reader engagement.
Response 2: Thank you. It is not clear from this comment where the redundancy occurs. Glucose modulating CSD is a common theme through the review and a mechanistic link between several factors. We would be happy to make revisions if you feel specific sections of the manuscript are repetitive.
Comment 3: It is highly recommended to change the title of the manuscript, as it mainly focuses on the relationship between CSD and glycemic states.
Response 3: Thank you for your suggestion. The impact of glycemic states is certainly a central thread of the manuscript. We have elected to modify the title to: Metabolic Pathophysiology of Cortical Spreading Depression: A Review
Comment 4: The text makes broad claims about relationships between migraines, glucose dysregulation, and CSD without citing specific studies or evidence. Including references to relevant literature would strengthen the arguments and provide a foundation for the claims made (Sleep Med Rev. 2021 Jan 29;59:101449).
Response 4: Thank you for this comment. We would be happy to revise any specific sections or claims that you feel are unsupported. However, we have elected to make no changes at this time as we feel that our claims are adequately supported by the included references. Your suggested citation does contain some relevant information, but it focuses on the interplay between glucose (or more accurately glycogen), sleep disorders, and migraine/depression and we do not feel it warrants inclusion.
Comment 5: Additionally, while it highlights many relationships, it could more explicitly outline gaps in current research and the implications of these findings for clinical practice.
Response 5: Currently the implications for clinical practice are unclear. There are numerous gaps in the literature that are beyond the scope of this paper. We feel, as stated in our Future Directions section, that the most immediate avenue for research lies in elucidating the pathways/roles of the various proteins and biomarkers that are induced by CSD.

Round 2
Reviewer 1 Report
Comments and Suggestions for Authors
Thank you for addressing my suggestions